# Comprehensive Analysis of Genomic Alterations in Hepatoid Adenocarcinoma of the Stomach and Identification of Clinically Actionable Alterations

**DOI:** 10.3390/cancers14163849

**Published:** 2022-08-09

**Authors:** Rongjie Zhao, Hongshen Li, Weiting Ge, Xiuming Zhu, Liang Zhu, Xiangbo Wan, Guanglan Wang, Hongming Pan, Jie Lu, Weidong Han

**Affiliations:** 1Department of Medical Oncology, Sir Run Run Shaw Hospital, College of Medicine, Zhejiang University, Hangzhou 310016, China; 2Cancer Institute, The Second Affiliated Hospital, College of Medicine, Zhejiang University, Hangzhou 310005, China; 3Department of Medical Oncology, Zhejiang Provincial People’s Hospital, Hangzhou Medical College, Hangzhou 314408, China; 4Department of Pathology, Cancer Hospital of the University of Chinese Academy of Sciences, Hangzhou 310005, China; 5Department of Radical Oncology, The Sixth Affiliated Hospital of Sun Yat-sen University, Guangzhou 518052, China; 6Department of Pathology, Sir Run Run Shaw Hospital, College of Medicine, Zhejiang University, Hangzhou 310016, China; 7Department of Gastroenterology, Gongli Hospital of Shanghai Pudong New Area, Shanghai University, Shanghai 200135, China; 8Department of Gastroenterology, The Tenth People’s Hospital of Tongji University, Shanghai 311202, China

**Keywords:** hepatoid adenocarcinoma of the stomach, whole-exome sequencing, mutagenesis mechanisms, aggressive behaviors, clinically actionable alterations

## Abstract

**Simple Summary:**

Hepatoid adenocarcinoma of the stomach (HAS) is a subset of gastric cancer (GC) histologically characterized by hepatocellular carcinoma-like foci with or without alpha-fetoprotein secretion, which is easily misdiagnosed. Genomic alterations and potential targets for this population are still largely unknown. Additionally, treatment regimens of HAS are mainly based on GC guidelines, which is not reasonable for diseases with great heterogeneity. The present study comprehensively depicts the genomic features of HAS, and they are significantly different from GC, AFP-producing GC (AFPGC), and liver hepatocellular carcinoma (LIHC). Multiple aggressive behavior-related amplificated or deleted regions in HAS are firstly reported. Moreover, reliable and practicable clinically actionable alterations for HAS are identified, providing evidence for making personalized therapy based on the genomic characteristics of HAS instead of GC.

**Abstract:**

Hepatoid adenocarcinoma of the stomach (HAS) is a rare malignancy with aggressive biological behavior. This study aimed to compare the genetic landscape of HAS with liver hepatocellular carcinoma (LIHC), gastric cancer (GC), and AFP-producing GC (AFPGC) and identify clinically actionable alterations. Thirty-eight cases of HAS were collected for whole-exome sequencing. Significantly mutated genes were identified. *TP53* was the most frequently mutated gene (66%). Hypoxia, *TNF-α*/*NFκB*, mitotic spindle assembly, DNA repair, and *p53* signaling pathways mutated frequently. Mutagenesis mechanisms in HAS were associated with spontaneous or enzymatic deamination of 5-methylcytosine to thymine and defective homologous recombination-related DNA damage repair. However, LIHC was characteristic of exposure to aflatoxin and aristolochic acid. The copy number variants (CNVs) in HAS was significantly different compared to LIHC, GC, and AFPGC. Aggressive behavior-related CNVs were identified, including local vascular invasion, advanced stages, and adverse prognosis. In 55.26% of HAS patients there existed at least one clinically actionable alteration, including *ERBB2*, *FGFR1*, *CDK4*, *EGFR*, *MET*, and *MDM2* amplifications and *BRCA1/2* mutations. *MDM2* amplification with functional *TP53* was detected in 5% of HAS patients, which was proved sensitive to *MDM2* inhibitors. A total of 10.53% of HAS patients harbored TMB > 10 muts/Mb. These findings improve our understanding of the genomic features of HAS and provide potential therapeutic targets.

## 1. Introduction

Hepatoid adenocarcinoma of the stomach (HAS) is a subset of gastric cancer (GC) histologically characterized by hepatocellular carcinoma-like foci with or without alpha-fetoprotein (AFP) secretion [1]. It is a rare cancer and has not been included in the four recently defined GC subtypes by The Cancer Genome Atlas Research Network based on molecular evaluation: EBV(+) tumors, chromosomal instability tumors, microsatellite unstable tumors, and genomically stable tumors [2]. However, previous studies reported that HAS was more aggressive in biological behavior than ordinary GC, including a higher rate of vascular invasion, regional and distant metastasis, and a strikingly worse prognosis [3,4]. Additionally, hepatocellular carcinoma-like foci in the stomach were easily misdiagnosed as metastatic liver hepatocellular carcinoma (LIHC), and wrong diagnosis and treatment decisions imposed a huge physical and economic burden on patients.

The mechanisms involved in the progression and clinically actionable targets of HAS are still largely unclear. The stomach and liver share a common embryonic origin from the foregut, making genetic progression and divergence critical factors in the development of HAS [5,6,7]. Based on immunohistochemical results, Shinichi Tsuruta MD. et al. [8] concluded that compared to GC, HAS had a relatively higher ratio of *ERBB2* expression (score 3+/2+: 21%/19%), as well as overexpression of *p16* and *IMP3*. Additionally, there were also studies showing that *TP53* was highly mutated and its abnormal function might affect HAS differentiation [1,9]. The copy number gains (CNGs) at 20q11.21–13.12 were linked to adverse biological behavior, but it was not statistically significant, which might be associated with limited information provided by panel-based sequencing technology. Therefore, it was worthwhile to further comprehensively depict the genomic characteristics of HAS by larger panel and explore the association with clinicopathological features and prognosis.

Moreover, presently, the treatment of HAS is based on GC guidelines as there is no specific and systemic treatment regimen for HAS patients. However, the morphology, biological behavior, and prognosis between HAS and GC are quite different, making it necessary to identify specific therapeutic targets. In addition, the predictive markers of immunotherapy efficacy in HAS, such as tumor mutation burden (TMB) and microsatellite instability (MSI), should be studied. The present study aimed to comprehensively compare the genomic alterations in HAS with those in LIHC, GC, and AFP-producing GC (AFPGC), and identify clinically actionable alterations.

## 2. Materials and Methods

### 2.1. Patients and Samples

This study focused on 38 cases of HAS and matched normal mucosa samples, of which 15 were collected between 2012 and 2017 from Sir Run Run Shaw Hospital, The First Affiliated Hospital of Zhejiang University, The Second Affiliated Hospital of Zhejiang University, Zhejiang Cancer Hospital, Zhejiang Provincial People’s Hospital, Shanghai Tenth People’s Hospital, and The First Affiliated Hospital of Sun Yat-sen University. The study was approved by the Ethics Committee of the affiliated Sir Run Run Shaw Hospital, School of Medicine, Zhejiang University. The other 23 cases of HAS with effective WES data were obtained from the dataset ERP128950 [10], and 31 cases of nonhepatoid differentiated patients were included as the AFPGC cohort. Additionally, we collected genomic alteration data of LIHC (364 cases) and GC (437 cases) from the TCGA database.

### 2.2. DNA Extraction and Library Construction

The genomic DNA was extracted from frozen tissues using the DNeasy Blood & Tissue Kits (Qiagen, Valencia, CA, USA). Degradation and contamination were evaluated on 1% agarose gel, and the concentration was measured using Qubit^®^ DNA Assay Kit in a Qubit^®^ 2.0 Fluorometer (Life Technologies, Carlsbad, CA, USA).

A paired-end 150 bp DNA library was constructed based on the manufacturer’s instructions (Agilent, Cedar Creek, TX, USA). Whole-exome capture was performed using Agilent SureSelect Human All Exon V6 kit (Agilent Technologies), and the captured libraries were sequenced on the Illumina Hiseq X ten platform. We then performed data quality control using fastp v0.20.1 with default parameters, including the removal of adapter contamination and low-quality reads. All the following bioinformatics were conducted on high-quality clean data. The sequencing coverage and quality statistics of each sample were summarized (Appendix A).

### 2.3. Single-Nucleotide Variants (SNVs) and Copy Number Variants (CNVs) Calling

We then aligned reads to the reference genome (hg38) using Burrows–Wheeler Aligner (BWA v0.7.17). Following the best practice workflow of GATK (v4.2), we performed base recalibration with the known sites of dbSNP and 1000 Genomes, orientation bias filter, cross-sample contamination estimation, germline population filter with gnomAD, and a panel of normals to reduce systematic artifacts of sequencing and data processing. Eventually, SNVs and INDELs with the flag of “PASS” were extracted for downstream analysis, and variants were annotated by VEP (v104.3). Enrichment analysis of hallmark pathways was conducted with all mutations recorded in maf file by the OncogenicPathways function with default parameters in the maftools package. Additionally, frequency matrices generated by single 5′ and 3′ bases flanking the mutated site and NMF decomposition were conducted for de novo signature analysis by the trinucleotideMatrix and extractSignatures functions in the maftools package. Significantly mutated genes (SMG) were identified by MutSigCV (v1.41).

Copy number variant calling was executed in CNVkit (v0.9.9) with default parameters. CNV was classified into five levels, including deep deletion (CN = 0), shallow deletion (CN = 1), diploid (CN = 2), low-level amplification (CN = 3), and high-level amplification (CN ≥ 4). Loss of heterozygosity (LOH) was inferred with allelic frequency of germline heterozygous SNP by CNVkit. Significantly amplified and deleted genome regions were identified in GISTIC (v2.0). The definition of amplificated or deleted regions was as follows: deep deletion (CN = −2), shallow deletion (CN= −1), diploid (CN = 0), low-level amplification (CN = 1), and high-level amplification (CN = 2). The overall survival (OS) analysis of CNV was performed in amplification (high-level and low-level) vs. non-amplification, and deletion (deep and shallow) vs. nondeletion.

Based on the potentially drug-related targets from the oncology knowledge base (OncoKB; https://www.oncokb.org (accessed on 13 February 2022)), we systematically evaluated the landscape of clinically actionable alterations in HAS based on the results of SNVs, INDELS, high-level amplifications, and deep deletions. Microsatellite instability (MSI) status was evaluated by MANTIS (v1.0.5).

### 2.4. Statistical Analyses

Categorical variables were compared using the Chi-Squared test or Fisher’s exact test, while continuous variables were compared using the Student’s *t*-test. Overall survival analysis was calculated using the Kaplan–Meier method with log-rank test. Statistical analyses were performed using R software v4.1.1 or Graphpad Prism v7.00. *p* < 0.05 was considered statistically significant.

## 3. Results

### 3.1. Clinicopathological Characteristics of HAS

A total of 38 patients were enrolled in this study. Nearly half of the tumors were located in the antrum and were categorized as stage III-IV. HAS was typical of local vascular invasion (71.05%) and liver metastasis (21.05%). Serum AFP increase was observed in 84.21% of the patients (Appendix A). The pathological features of HAS were re-evaluated by HE-stained sections under the light microscope. Different from normal gastric mucosa cells arranged in a glandular tube, hepatoid-differentiated cells had an irregular shape and were mitotically active (Appendix A). They were arranged in various patterns, including nest, trabeculae, and glandular tubules (Appendix A), accompanied by a tendency to infiltrate around, including local vascular invasion and liver metastasis (Appendix A). Immunohistochemical results showed that 86.67% of cases were positive for AFP (Appendix A).

### 3.2. HAS Was Distinct in the Somatic Mutation Landscape

To compare the genomic heterogeneity of HAS, LIHC, GC, and AFPGC, we analyzed somatic mutation profiles and identified 7313 somatic alterations for 5154 genes in HAS, and the mutation proportion of frame-shift, nonsense, nonstop, and splice site was approximately 13.5% compared with 14.7% in LIHC, 22.1% in GC, and 13.0% in AFPGC, which could damage the protein structure and result in loss-of-function.

The SMGs with *p* < 0.001 were shown in the oncoplot (Figure 1A, Appendix A). *TP53* was the most recurrent gene (66%) in HAS, mutating more frequently than in LIHC (29%, *p* = 1.33 × 10^−5^), but similar to GC (47%, *p* = 0.028) and AFPGC (55%, *p* = 0.46) (Figure 1B and Appendix A). Additionally, compared with LIHC, mutation frequencies of *CCDC15*, *CMEB2*, *PHLDA1*, *PTH2*, *PHKA2*, *IRF2BP2*, *C2orf44*, *NFYA*, and *SP8* were significantly higher in HAS (*p* < 0.0167). Nine of the thirteen SMGs (*CCDC15*, *GMEB2*, *PHLDA1*, *PTH2*, and *NFYA*) had different mutation frequencies in HAS and GC (*p* < 0.0167). Intriguingly, only one of them (*PHLDA1*) had a differentially mutated tendency in HAS and AFPGC (*p* = 0.060), indicating that HAS had more genomic differences with LIHC and GC, and shared mutational characteristics with AFPGC.

*TP53* was the only common SMG in the four cancers (Figure 1C). Further analysis showed that most of the mutation sites were located in the *p53* DNA-binding domain (Figure 1D), similar to LIHC, GC, and AFPGC. Notably, the distribution of hotspot mutations was different. Unlike missense mutation in codon 249 for LIHC, missense mutation in codon 175 and 273 for GC, and splice site mutation in codon 332 for AFPGC, the most common type for HAS was a nonsense mutation in codon 91, which could result in loss-of-function of *TP53* by reducing the expression of the protein. Protein structure changes varied widely due to different mutation types and sites, affecting the disease outcomes.

### 3.3. Mutated Pathway Analyses in HAS

Pathway enrichment analysis enhanced our understanding of the biological effects of mutations. Based on 50 hallmark gene sets from GSEA (Gene Set Enrichment Analysis), we found that multiple oncogenic pathways were frequently mutated in HAS, and part of them were distinct from those in LIHC and GC (Figure 2A,B) [11]. Mitotic spindle assembly was the most recurrently affected pathway in HAS, where 89.5% of cases (*n* = 34) had at least one mutation in this pathway, including *MYH9* (*n* = 4, 10.5%), *CDC27* (*n* = 4, 10.5%), and *TRIO* (*n* = 4, 10.5%). Other recurrently and differentially mutated pathways in HAS vs. GC and HAS vs. LIHC were as follows: hypoxia pathway with 51 mutant genes in 30 HAS patients (78.9% vs. 47.5% of LIHC, *p* < 0.001; 78.9% vs. 59.5% of GC, *p* < 0.05), *TNF-α/NF-kB* pathway with 48 mutant genes in 28 HAS patients (73.7% vs. 45.9% of LIHC, *p* = 0.001; 73.7% vs. 55.1% of GC, *p* = 0.028), DNA repair pathway with 38 mutant genes in 29 HAS patients (76.3% vs. 50.0% of LIHC, *p* = 0.002; 76.3% vs. 70.0% of GC, *p* > 0.05), and *p53* pathway with 42 mutant genes in 31 HAS patients (81.6% vs. 62.9% of LIHC, *p* = 0.021; 81.6% vs. 79.4% of GC, *p* > 0.05). However, pathway mutation frequencies between HAS and AFPGC only showed a different tendency (*p* > 0.05; Figure 2C).

*TP53* mutation was observed in 66% of HAS patients, and it was the common node of *p53*, Wnt/β-Catenin, and the DNA-repair pathway indicating that these pathways might function simultaneously in the development and progression of HAS (Appendix A). Moreover, pairwise associations between recurrent alterations were used to determine significant mutual exclusivity and co-occurrence, which identified the co-mutations of *LAMA1* in the epithelial–mesenchymal transition (EMT) pathway and *PHLDA1* in the *TNF-α*/*NF-κB* pathway, and the co-mutations of *PKHD1* in the apical surface pathway and *DMD* in the G2M checkpoint pathway, indicating that many aberrant pathways tend to co-exist. We also found that multiple genes in a pathway simultaneously mutated in a patient, such as *LAMA1*, *LAMA2*, and *COL4A2* in the EMT pathway (Figure 2D and Appendix A).

### 3.4. Mutagenesis Mechanism Heterogeneities among HAS, LIHC, GC, and AFPGC

Different mutational mechanisms could induce specific combinations of mutation types. To characterize such kinds of mutational signatures, we calculated the distribution of six single-base substitutions, including C > A, C > G, C > T, T > A, T > C, and T > G, and found that the dominant substitution pattern among patients with HAS was C > T changes (57.67% of all SNVs), followed by C > A (12.33%), C > G (10.14%), T > G (7.44%), T > C (7.33%), and T > A (5.09%) (Figure 3A), which was quite similar to that in GC and AFPGC. In contrast, in LIHC, the proportions of these base substitutions were relatively evenly distributed (Appendix A).

Further analysis of the context of bases immediately 3′ and 5′ to the mutated base generated a total of 96 different mutation patterns with the above six types of substitutions, and we identified six mutation signatures in HAS patients and calculated their correlations with known mutation signatures from the Catalogue of Somatic Mutations in Cancer (COSMIC) database (v3; Figure 3B) [12]. The first signature was correlated with the SBS2 COSMIC signature (cosine similarity = 0.664), *APOBEC*-induced mutagenesis, which was typical of enrichment of C > T or C > G changes at TCW motifs, in which W was A or T (Figure 3C) and was involved in the local hypermutation phenomenon. The second signature showed a high cosine similarity with the SBS1, which was associated with the endogenous mutation process of spontaneous or enzymatic deamination of 5-methylcytosine to thymine, and served as a cell division/mitotic clock that controls division rates (cosine similarity = 0.904). The fourth signature was correlated with the SBS6 (defective DNA mismatch repair, cosine similarity = 0.768), and the patients with this mutation suffered a strikingly higher mutation load than the other patients (Figure 3D), which might be associated with increased genomic instability. Similarly, the fifth signature had a high correlation with defective homologous recombination-related DNA damage repair (cosine similarity = 0.846; Figure 3C). The aetiology in the third and sixth signatures (SBS17b and SBS5) was still unclear and remained to be further annotated by COSMIC.

All these SBS COSMIC signatures in HAS except SBS2 were shared in AFPGC, and most of them were associated with defective DNA repair indicating the instable genomic features in AFPGC (Appendix A). In GC, the mutagenesis mechanisms in half of the patients (48.3%) were characterized by polymerase epsilon exonuclease domain mutations and spontaneous or enzymatic deamination of 5-methylcytosine to thymine (Appendix A). In contrast, exposure to aflatoxin and aristolochic acid (cosine similarity > 0.9) was considered one of the main factors in LIHC (Appendix A), which was consistent with the current reports on the causes of liver cancer [13,14].

### 3.5. Copy Number Variants Correlated with Outcomes of HAS

We then evaluated the copy number variants in HAS and found extensive CNVs, including 22 significant amplifications and 50 deletions (*q*-value < 0.05; Figure 4A). Regions such as chr4q35.2, chr5q35.3, chr8q24.21, chr9p21.3, and chr19p13.3 were frequently altered in over 70% of patients. We exclusively focused on the aggressive biological behavior of HAS and analyzed its correlation with CNVs. Deletions in chr7q22.1 and chr8p23.3, and amplification in chr10q21.2 were positively associated with local vascular invasion or liver metastasis, while deletions in chr6q26, chr9p11.2, chr13q11, chr22q11.21, chr22q13.33, and amplifications in chr8p23.1 were negatively associated with local vascular invasion, liver metastasis, or advanced stages (*p* < 0.05; Figure 4B). Additionally, patients with amplifications in chr8q21.2, or deletions in chr1q44 and chr1p36.21, had shorter overall survival, while deletions in chr5q13.2, chr17q12, chr17p11.2, and chr22q13.33 had better overall survival (*p* < 0.05; Figure 4C). Additionally, some variants showed potential correlations with these adverse outcomes (*p* > 0.05; Appendix A). Similar to HAS, LIHC, GC, and AFPGC also harbored several significantly altered regions (Appendix A), and six of them (amplifications in chr8q24.21, chr11q13.3, and chr19q12, and deletions in chr1p36.11, chr14q32.33, and chr19p13.3) were commonly shared by the four types of cancers (Appendix A), suggesting that these variants might be an early molecular event [15,16].

CNVs in oncogenes and tumor suppressor genes (TSGs) could cause cell growth disorder and contribute to the aberrant proliferation of tumor cells. In the COSMIC gene list, we identified a total of six oncogenes and 14 TSGs located in these frequently altered regions (Figure 4A). OS analysis of all oncogenes and TSGs recorded in the COSMIC database showed that deletions of four TSGs (*SBDS*, *FANCC*, *PTCH1*, and *XPA*; Appendix A) and amplifications of six oncogenes (*PIK3CB*, *GATA2*, *WWTR1*, *MUC4*, *RARA*, and *AFF3*; Appendix A) predicted a shorter OS (*p* < 0.05). Additionally, deletions of the left four TSGs (PPP6C, WNK2, CASP3, and CD274; Appendix A) and amplifications of the thirteen oncogenes (*BCL6*, *ETV5*, *LPP*, *SOX2*, *ERBB2*, *MECOM*, *FOXR1*, *HEY1*, *CCR7*, *IL7R*, *MYC*, *TRRAP*, and *CTNNA2*; Appendix A) were potentially associated with a shorter OS (*p* > 0.05).

### 3.6. Overview of Clinically Actionable Alterations

Clinically actionable alterations analysis showed that a total of 17 targets from 21 patients (55.26%) were identified (Figure 5A). Specifically, 17 patients (44.74%) had no clinically actionable alteration, 14 (36.84%) had one, and 7 (18.42%) had at least two targets. *ERBB2* amplification was the most recurrent clinically actionable alteration, followed by amplifications in *MDM2*, *FGFR1*, *MET*, *CDK4,* and *EGFR*, and oncogenic mutations in genes such as *BRCA1/2*, *NF1,* and *ALK*. Clinically, several *ERBB2*-amplified GC patients could benefit from trastuzumab target therapy. Therefore, we were greatly interested in the higher proportion of *ERBB2* amplification in HAS (21.05%), while it was approximately 14% in GC, 0.6% in LIHC, and 25.8% in AFPGC, indicating that anti-*ERBB2* treatment in HAS is also a promising option (Appendix A). Additionally, *MDM2* amplification with functional *TP53* was found for the first time in HAS patients. Multiple clinical trials showed that solid tumors with *MDM2* amplification as well as wild-type or functional *TP53* were sensitive to milademetan and ALRN-6924 [17,18,19], an inhibitor of *MDM2*, suggesting that HAS patients might also benefit from the therapy since the proportion of the genotype added up to 5% (Figure 5B; Appendix A).

Increasing evidence showed that TMB and MSI status were potential biomarkers for predicting patients’ response to the PD-1/PD-L1 inhibitors in various cancers [20,21]. However, the therapeutic value of TMB and MSI in HAS has not been reported before. In this study, TMB of HAS ranked fifth across 33 types of cancers from TCGA cohorts with a median TMB of 3.41 muts/Mb (range: 0.22–69.28 muts/Mb; Appendix A), indicating obvious heterogeneity among patients. Four HAS patients harbored TMB > 10 muts/Mb, which might be sensitive to immunotherapy. Notably, TMB in HAS and AFPGC was significantly higher than LIHC (range: 0.03–34.16 muts/Mb; *p* < 0.001). There were no significant differences in HAS vs. GC (range: 0.03–268.66 muts/Mb; *p* = 0.53) and HAS vs. AFPGC (range: 0–30.88 muts/Mb; *p* = 0.97). Microsatellite instability analysis showed all HAS patients were microsatellite stable (MSS; Appendix A).

## 4. Discussion

HAS is a kind of tumor located in the stomach and typical of hepatoid differentiation with or without the secretion of AFP, which has a strong ability of invasion and metastasis [3,4]. Currently, genomic alterations and potential treatment targets for HAS are still largely unknown due to the rarity of these cases. In the present study, in more than half of HAS cases there existed local vascular invasion, and 21.05% of HAS cases metastasized to the liver, turning into advanced stages. The genetic landscape showed that HAS was obviously different compared to LIHC and GC in terms of recurrent SNVs, pathways, mutagenesis mechanisms, and CNVs, but shared mutational characteristics with AFPGC to some extent. Notably, multiple significantly amplified or deleted genome regions were firstly found involved in local vascular invasion, liver metastasis, advanced stages, and overall survival of HAS, which could enhance the understanding of aggressive biological behaviors. A total of 55.26% of HAS patients harbored at least one clinically actionable alteration, providing evidence for personalized treatment.

Ji et al. depicted the recurrently mutated genes and regions in HAS, but the correlations of these variants with the aggressive behaviors of HAS such as invasion, metastasis, stages, and adverse prognosis were not researched [9]. In addition, due to the overlap in morphology and location of HAS, GC, LIHC, and AFPGC, we made a comprehensive comparison of genomic characteristics, and it improved our knowledge about HAS and was helpful to further studies.

Among the SMGs, *TP53* mutation was the most frequent variant. It was the shared node in *p53*, *Wnt*/*β-Catenin*, and the DNA-repair pathway, indicating that multiple oncogenic pathways co-exist and function simultaneously in most HAS patients, which might promote tumorigenesis and invasion of HAS [22,23]. Additionally, six of the SMGs were more highly mutated in HAS than in both LIHC and GC, including *PHLDA1*, *GMEB2*, *CCDC15*, *PTH2*, *NFYA*, and *C2orf44*. Mutation frequencies of genes mentioned above were similar in HAS and AFPGC except *PHLDA1* (*p* = 0.060), high expression of which could inhibit the survival and metastasis of gastric cancer and served as a TSG [24]. *PHLDA1* downregulation was proved to promote acquisition and maintenance of drugs resistance targeting receptor tyrosine kinase, which might suppress the effect of trastuzumab treatment on *ERBB2* amplification cancers [25].

The mutation spectrum in the context of 96 types of base constitution varied a lot, and the pattern could reflect mutagenic mechanisms in tumorigenesis. The deficiency of DNA mismatch repair was common in all the cancers, but the preference for the TCW mutation pattern mediated by *APOBEC* was only identified in HAS, indicating genomic specificity [26]. For LIHC, chemical carcinogenesis was considered the main mutation initiation factor, including exposure to aflatoxin and aristolochic acid [27], which have been widely accepted and ensure the reliability of the results.

Increasing evidence reveals that CNVs induced by various endogenous and exogenous stimuli existed in tumor tissues and were critical in genetic heterogeneity [28], transformation of precancerous lesions [29], and treatment efficacy [30]. There were six mutated regions detected in all of these cancers, including amplifications in chr8q24.21, chr11q13.3, and chr19q12, and deletions in chr1p36.11, chr14q32.33, and chr19p13.3, which might play an important role in cancer development and progression. For example, chr8q24.21 amplification was frequent in advanced GC [31], and involved in liver metastasis of rectal adenocarcinoma [32]. chr19q12 amplification was prevalent in multiple tumor types [33]. Consistently, in 65.8% of HAS cases there existed the amplification of chr19q12. Amplification of the *CCNE1* locus in this region was synthetic lethal related, targeted by *PKMYT1* kinase inhibition, and it was also a promising target in AFPGC, which warranted further research in HAS [10,33]. Deleted regions-associated studies were mainly involved in gastrointestinal defects, Peutz-Jeghers syndrome (19p13.3), and immunodeficiency (14q32.33) [34,35,36,37]. There were 32 regions specifically amplificated or deleted in HAS (Appendix A), and the amplification of chr22q11.21 and deletions of chr1p36.21, chr7q11.21, chr7q22.1, chr9q12 as well as chr12q24.33 predicted a tendency of shorter OS. Intriguingly, Farshidfar et al. demonstrated that chr22q11.21 amplification in melanoma was strongly associated with inferior survival, metastasis, and downregulation of immunotherapy response-related genes [38], indicating its important role in HAS. Additionally, chr10q21.2 amplification was linked with precursor B-cell acute lymphoblastic leukemia of childhood [39], and in this study we found it was associated with local vascular invasion. These results provided insights into the mechanisms underlying aggressive behaviors.

The mutation landscape of HAS was significantly distinct from GC, including SNVs, mutational patterns, pathways, and CNVs. Therefore, it was necessary to understand HAS comprehensively and develop treatment strategies based on its genomic features. The previous study considered MAT2A and AHCY as potential targets to HAS by clustering and comparing the transcription profiles of HAS and TCGA-GC. However, there must be batch effects between two groups from two different cohorts, which were not corrected and might notably affect the reliability of the result [9]. We identified targets of four levels which were more reliable and practicable, including FDA-approved drugs, standard card, clinical evidence, and biological evidence [40]. Our data showed that 55.26% of HAS patients carried at least one potentially targetable alteration. *ERBB2* amplification was present in more than 20% of HAS patients who had a shorter OS tendency (Appendix A), which was similar to GC [41]. However, it created novel agents and opportunities. *ERBB2*-targeted therapy greatly improved the prognosis of GC patients with *ERBB2* amplification [42,43]. Such benefits were also observed beyond gastric and breast cancer, including in bladder, biliary, and colon cancers [43]. Therefore, it might also serve as a promising treatment choice for HAS patients. Additionally, *MDM2* was a negative regulator of the tumor suppressor *TP53* [44]. Inhibiting the interaction between wild-type *p53* and *MDM2* was one of the main strategies to develop compounds targeting dysfunctional *p53* for cancer treatment [45,46], which has been introduced into clinical trials, showing that solid tumor patients with *MDM2* amplification as well as wild-type or functional *TP53* could benefit from *MDM2* inhibitors, milademetan, and ALRN-6924 [17,18,19,47]. The proportion of the genotype added up to 5% in HAS patients indicating an *MDM2* inhibitor is also a promising option.

Additionally, mutations in *BRCA1* (3%) and *BRCA2* (8%) were detected in HAS. An increasing number of clinical trials have shown that various malignancies with *BRCA* mutation may potentially be sensitive to PARP inhibitors [48], which offered a potential treatment option for HAS. High TMB and high microsatellite instability are conventionally regarded as useful biomarkers to predict immunotherapy response. At present, the definition of high and low levels of TMB is still controversial, but the most widely accepted standard is 10 muts/Mb [49]. In the present study, 10.52% of HAS patients harbored high TMB, and the proportions in TCGA-GC, TCGA-LIHC, and AFPGC cohorts were 21.5%, 1.6%, and 3.2% respectively. All HAS patients were in MSS condition, suggesting that only a small number of HAS patients might benefit from immunotherapy. This requires further validation.

There was a limitation in our study. The incidence of HAS was very low, resulting in a small sample size, which may lead to some deviations in the results. Some novel variations related to the malignant characteristics of HAS were identified. Further functional validation of the variations may have improved the reliability of the conclusion, but the scarcity of samples made the experimental verification difficult.

## 5. Conclusions

HAS has its specific molecular variant characteristics, and it is different from LIHC, GC, and AFPGC. We have identified progression-associated alterations, and provided an overview of clinically actionable alterations of HAS, including *ERBB2* amplification and *MDM2* amplification with functional *TP53*, which are helpful for establishing a treatment scheme based on its own genetic background.

## Figures and Tables

**Figure 1 cancers-14-03849-f001:**
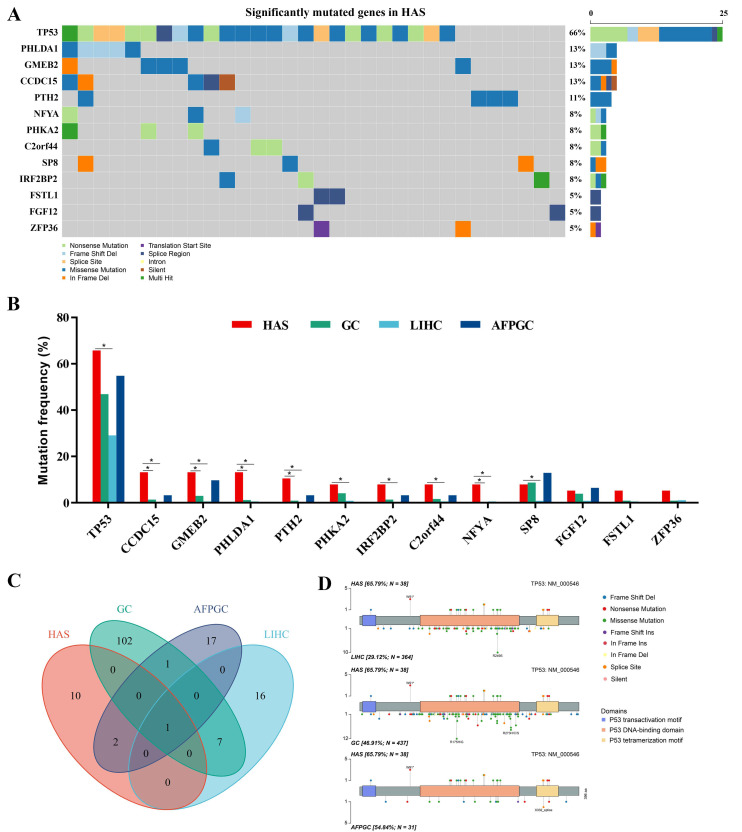
**Mutational landscape of hepatoid adenocarcinoma of the stomach (HAS).** (**A**) The significantly mutated genes (SMGs) in 38 HAS patients. Different variant types are highlighted in different colors. *p* < 0.001 is statistically significant. (**B**) Comparison of frequencies of SMGs in HAS with liver hepatocellular carcinoma (LIHC), gastric cancer (GC), and AFP-producing GC (AFPGC). ** p* < 0.0167, statistically significant. (**C**) Intersection of all SMGs in these cancers. (**D**) Somatic mutation site distributions of *TP53*.

**Figure 2 cancers-14-03849-f002:**
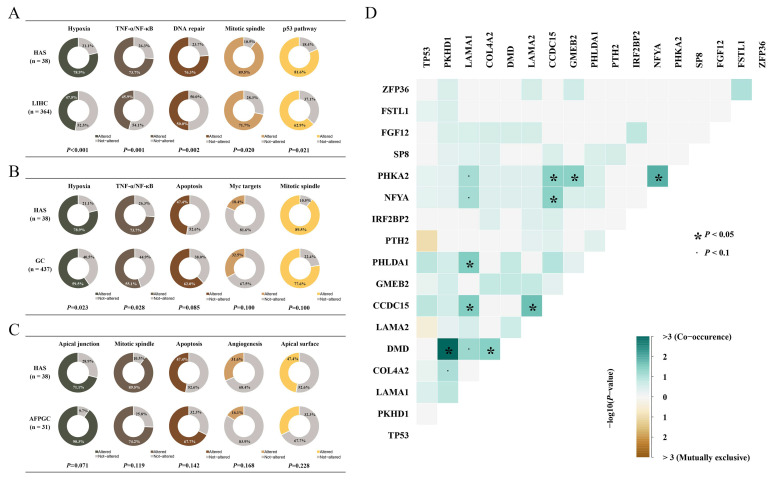
**The proportion of mutated pathways.** The proportion of patients with mutated genes in the described pathways and its differences in (**A**) HAS vs. LIHC, (**B**) HAS vs. GC, and (**C**) HAS vs. AFPGC. The curated gene sets for each pathway are obtained from the GSEA database. (**D**) Co-occurrence or mutual exclusivity patterns for the SMGs and representative genes. *p* < 0.05 is statistically significant.

**Figure 3 cancers-14-03849-f003:**
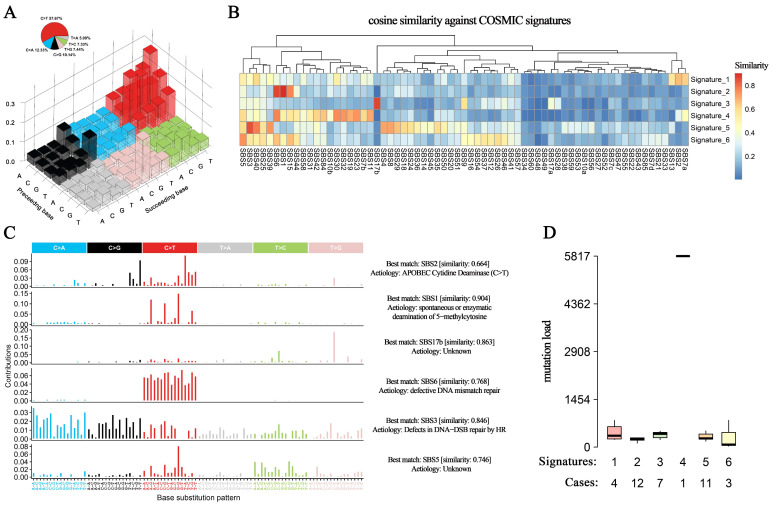
**Base substitution patterns and mutation signatures.** (**A**) A total of 96 types of base substitution patterns and their somatic mutation proportion are derived from HAS. Red: C > T, blue: C > A, black: C > G, pink: T > G, green: T > C, and grey: T > A. The height of the columns represents the proportion of each base substitution. (**B**) Six signatures inferred from these patterns are mapped to known signatures in the COSMIC database to calculate cosine similarities. (**C**) Annotations to signatures in HAS indicate potential mutagenesis mechanisms. (**D**) The number of patients and mutation load in each signature. Colors represent different signatures. N represents the number of patients attributed to each signature.

**Figure 4 cancers-14-03849-f004:**
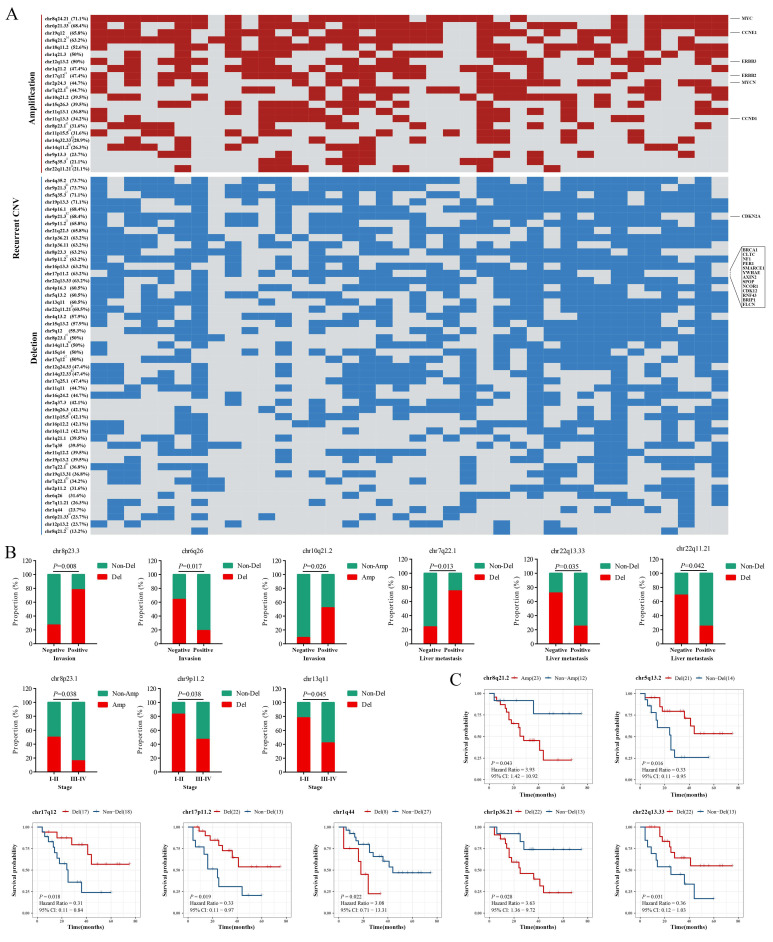
**Copy number variants (CNVs) in HAS.** (**A**) The heat map shows the recurrent CNVs in HAS. Oncogenes and tumor suppressor genes (TSG) in these regions are marked at the right of the heat map, and the list is provided by the COSMIC database. Red: amplification, blue: deletion. The repeatedly occurred regions marked with a-l are located in different wide peak limits which were recorded in Appendix A. The percentage in each row represents the number of samples with the CNVs. Mutated regions are significantly related to (**B**) local vascular invasion, liver metastasis, and advanced stages and (**C**) prognosis. *p* < 0.05 is statistically significant.

**Figure 5 cancers-14-03849-f005:**
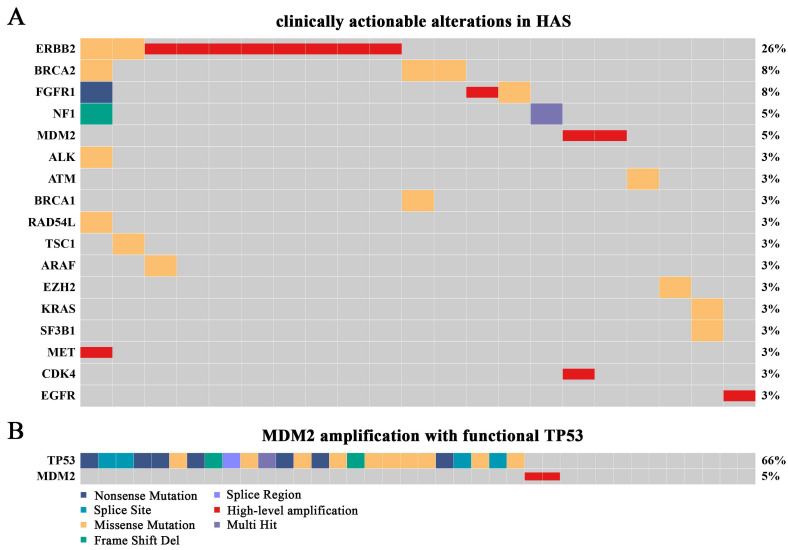
**Overview of clinically actionable alterations in HAS.** (**A**) The landscape of potentially targetable alterations based on the results of SNVs, INDELS, high-level amplifications, and deep deletions. Different variant types are highlighted in different colors. (**B**) Patients with high-level amplification of *MDM2* and functional *TP53*.

## Data Availability

The WES data generated in this study are available in the National Genomics Data Center under accession number HRA001552. The accessible link is as follows: https://ngdc.cncb.ac.cn/gsa-human/s/B4fS79Vl (accessed on 1 January 2022). Other data that support the findings of this study are available from the corresponding author upon request.

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
