# Peer review of "Comprehensive Analysis of Genomic Alterations in Hepatoid Adenocarcinoma of the Stomach and Identification of Clinically Actionable Alterations"

_cancers, 2022, doi:10.3390/cancers14163849_

Round 1

Reviewer 1 Report

This is a detailed and methodologically well-conducted work on the molecular landscape of the hepatoid adenocarcinoma of the stomach (HAS). 34 cases with HAS were studied with respect to genetic landscape. The study analyzes the detected gene variants and also ranks them in terms of clinical relevance.

It would further enhance the clinical relevance of the work if the  authors also would mentioned the CPS score of the four patients with a TMB > 10 muts/Mb.

Did the patients with TMB > 10 muts/Mb receive an immune checkpoint treatment and  received the patients with ERBB2 amplification a Her2 based antibody therapy?  If so, the clinical response would be interesting.

Overall, this is a high quality paper that can be fully recommended for publication.

Reviewer 2 Report

In this study, the authors characterized the genomic alterations of hepatoid adenocarcinoma of the stomach and compared with other subtypes of gastric cancer. The result is interesting, and the data can be a valuable resource for the field of gastric cancer research. However, the manuscript also suffers from multiple issues that need to be addressed in a revised version.

1.     The current “Materials and Methods” section is too brief, and the authors did not sufficiently describe their approach for several analyses. For example, for the pathway enrichment analysis, what was the tool used and what were the specific steps? Were only significantly mutated genes used as input?

2.     Similarly, for the mutation signature analysis, what was the tool or algorithm used? What was the number of mutations for the signature analysis for each sample? If the somatic mutations were too few, the accuracy can be questionable.

3.     Moreover, how were somatic mutations filtered after they were called by Mutect2? If not filtered properly, the analysis of identifying significantly mutated genes may be affected.

4.     For the detection of significantly mutated genes, using P < 0.05 can be too loose. The authors should use the Q value to account for the multiple testing, such as using Q < 0.1

5.     Also, for the comparison between different subtypes (HAS, GC, LIHC and AFPGC), the authors should also correct for multiple testing.

6.     What is the copy number status of TP53 across patients? For patients with TP53 mutation, is there concurrent loss of the WT allele (loss of heterozygosity). For patients without TP53 mutation, is there any with homozygous deletion

7.     For Figure 4A. What was the definition of amplification and deletion? Also regions such as 9p21.3 occurred more than once, which is confusing. Perhaps a GISTIC signal plot (with Q/p value and G-score) is more informative that what it is now.

8.     What are the purity and ploidy of these HAS tumor samples?

Round 2

Reviewer 2 Report

I do not have additional comments for the authors.